# Bacterial Pathogens Causing Pneumonia Post Hematopoietic Stem Cell Transplant: The Chronic GVHD Population

**DOI:** 10.3390/pathogens12050726

**Published:** 2023-05-17

**Authors:** Said Chaaban, Andrea Zimmer, Vijaya Raj Bhatt, Cynthia Schmidt, Ruxana T. Sadikot

**Affiliations:** 1VA Nebraska Western Iowa Health Care System, Omaha, NE 68105, USA; rsadikot@unmc.edu; 2Division of Pulmonary, Critical Care & Sleep, Department of Internal Medicine, University of Nebraska Medical Center, Omaha, NE 68198, USA; 3Division of Infectious Diseases, Department of Internal Medicine, University of Nebraska Medical Center, Omaha, NE 68198, USA; andreaj.zimmer@unmc.edu; 4Division of Hematology and Oncology, Department of Internal Medicine, University of Nebraska Medical Center, Omaha, NE 68198, USA; vijaya.bhatt@unmc.edu; 5McGoogan Health Sciences Library, University of Nebraska Medical Center, Omaha, NE 68198, USA; cmschmidt@unmc.edu

**Keywords:** bacterial pneumonia, chronic graft versus host disease, allogeneic hematopoietic stem cell transplant

## Abstract

Allogeneic stem cell transplantation is a lifesaving treatment for many malignancies. Post-transplant patients may suffer from graft versus host disease in the acute and/or the chronic form(s). Post-transplantation immune deficiency due to a variety of factors is a major cause of morbidity and mortality. Furthermore, immunosuppression can lead to alterations in host factors that predisposes these patients to infections. Although patients who receive stem cell transplant are at an increased risk of opportunistic pathogens, which include fungi and viruses, bacterial infections remain the most common cause of morbidity. Here, we review bacterial pathogens that lead to pneumonias specifically in the chronic GVHD population.

## 1. Introduction

Allogeneic hematopoietic stem cell transplantation (alloHSCT) is a lifesaving treatment for a multitude of benign and malignant diseases [1]. Annually, more than 50,000 alloHSCTs are performed worldwide [1,2]. Pulmonary complications remain a major contributor to morbidity and mortality following alloHSCT. Etiologies include non-infectious diseases, as well as lung infections caused by bacteria, fungi, and viruses [1].

Despite the utilization of antimicrobial prophylaxis and healthcare infection prevention measures, bacterial infections continue to cause significant morbidity and mortality in alloHSCT recipients [1,3]. Recent data report that up to 20–30% of alloHSCT recipients develop at least one episode of pneumonia, with bacteria being the predominant causative pathogen [3,4,5]. Increased susceptibility to bacterial organisms occurs due to alterations in the immune system, disruption of the microbial flora, lung architectural derangements, and malnutrition [3]. In addition, the frequent and prolonged exposures to healthcare systems increase the risk of acquiring nosocomial pathogens, including resistant bacteria [3].

The immunocompromised status following alloHSCT is multifactorial and affects multiple pathways of immune function. High-intensity cytotoxic conditioning chemotherapy is generally administered in the days prior to the infusion of allogeneic donor stem cells. This preparatory regimen functions both to eradicate any residual malignancy and to prevent native lymphocytes from attacking donor cells to optimize chances for successful engraftment. The preparative cytotoxic chemotherapy regimen targets rapidly dividing cells and, therefore, destroys hematopoietic cells and causes damage to mucosal barriers [6]. This disruption of oral, respiratory, and gut mucosa allow organisms to invade or relocate into underlying tissues. Recipients of alloHSCT are generally neutropenic for more than 10–14 days until engraftment of donor neutrophils, which places them at high risk for infections caused by bacteria and other pathogens. Furthermore, the recovery of lymphocyte cells and function can take months to years depending on cell source and iatrogenic immunosuppression post alloHSCT, causing prolonged deficiencies in cellular and humoral immunity [7]. Graft-versus-host-disease (GVHD) is a multisystem alloreactive inflammatory process by which donor lymphocytes recognize recipient tissue as “non-self” and can lead to significant multi-organ dysfunction. GVHD is a leading cause of morbidity and mortality in alloHSCT recipients and generally requires immunosuppression both prophylactically in the early months post alloHSCT, as well as for treatment of acute GVHD flares [8]. The depth and duration of this immunosuppression directly influences risk for opportunistic infections [9]. Risk for infection and GVHD post alloHSCT varies according to conditioning regimens, donor type (related versus unrelated), recipient traits (gender, age, and CMV serostatus), HLA match (matched, haploidentical, and mismatch), and cell source (peripheral blood, bone marrow, and umbilical cord), among other factors [10,11].

## 2. Chronic GVHD

Chronic graft versus host disease (cGVHD) is defined based on standard criteria defined by the National Institute of Health and is divided into a limited and extensive form [12]. The cGVHD population usually has dysfunctional cellular and humoral immunity, which is compounded by immunosuppressive agents used for its treatment [12,13]. The incidence of pneumonia declines by 100 days post alloHSCT with the exception of patients with cGVHD [1]. Nearly 28% of patients with cGVHD have three or more infections by 6 months post transplant [14]. Chronic GVHD causes inflammation, tissue injury, lymphoid organ dysfunction (including spleen, and thymus), dysregulated T and B cell responses, and abnormal tissue repair, often leading to fibrosis [14,15]. These complex processes result in cGVHD manifestations such as bronchiolitis or sclerodema and induce prolonged cellular and humoral immune deficits. Encapsulated bacteria, such as *Streptococcus pneumoniae* and *Haemophilus influenzae*, have been dominantly seen in this population [1,12,16,17,18]. Pneumonia in the chronic GVHD patient carries a fivefold risk for mortality [12]. We hereby present all other cases reported in the literature of bacterial pathogens causing pneumonia in the chronic GVHD population.

## 3. Pathogenesis of Bacterial Pneumonia

Bacteria reach the lung through inhalation, aspiration, migration from the proximal airway, or hematogenous spread [3]. The majority of the pathogens are generally expelled via the mucociliary process along with other particulates trapped in the viscous and elastic fluid that lines the airways. Bacteria need to breach normal barrier defenses to reach the lung periphery [3].

Both structural and immunologic barriers protect the lungs from entry of invading pathogens. In an immunocompetent host, these barriers are often effective in eliminating most infections [3]. However, the resulting immune reaction in response to infection causes tissue injury and systemic inflammation [3]. Pneumonia, as a syndrome, is a culmination of these responses. It constitutes the radiographic findings that happen as a result of airspace filled by edema, debris, and white blood cells along with the systemic response fever and leukocyte elevation along with a productive cough [3].

Cancer, along with its treatments, leads to changes in both the innate and adaptive responses to a bacterial pathogen [3]. In addition, functional and anatomical defects may arise either directly related to the underlying neoplasm or its associated therapy. Complications related to therapy may result in a need for hospitalization and invasive procedures, which increases risk for acquiring nosocomial pathogens [3]. Furthermore, due to impaired immune function, the clinical presentation or radiographic findings of pneumonia may be blunted, sometimes leading to delayed diagnosis [3].

Patients with GHVD can be uniquely susceptible to bacterial pneumonias secondary to chronic inflammation and damage to tissue within airways and lungs along with the chronic deficits in cell-mediated and humoral immunity [3]. Immunosuppression for active GVHD will only increase the risk of infection and susceptibility to bacteria [3].

## 4. Methods

We performed a systematized review with EMBASE (via embase.com, 1974-present version) and simultaneous MEDLINE and CINAHL searches (via EBSCOhost). Search strategies included subject headings and keywords for the three search concepts: (1) hematopoietic transplant, (2) chronic GVHD, and (3) bacterial pneumonia (see complete search strategies in the appendix).

Filters were used to remove conference abstracts from the EMBASE results and to split all search results into three groups: (1) articles indexed as case reports, (2) review articles not indexed as case reports, and (3) all articles not retrieved by the case report or review search. No language or publication date filters were applied. All result groups were added to the project’s EndNote database. Both EndNote and Zotero duplicate detection tools were used to identify duplicates. Results concerning pediatric cases were then separated by searching the EndNote database for records containing words beginning with “pediatric”, “paediatric”, “infan”, “neonat”, “newborn”, or “adolescen”, but not containing “adult”.

The 182 total EMBASE search results and the 125 total results of the simultaneous MEDLINE and CINAHL searches were imported into our review’s EndNote database. After removal of the 82 duplicate records identified by the EndNote and Zotero duplicate detection tools, 225 records for unique articles remained for the title abstract review. We chose the 40 that were most relevant and with no redundancy of information for inclusion in this review; cases of pneumonia due to *Streptococcus pneumoniae* or *Haemophilus influenzae* were excluded (Figure 1).

## 5. Mycobacterium Tuberculosis

The incidence of tuberculosis (TB) varies from 0.001% to more than 10% in highly endemic countries [19]. The incidence of active disease amongst alloHSCT recipients is nearly triple compared with autologous HSCT recipients, with the lungs being the most affected organ [1,19]. Patients with cGVHD are particularly susceptible given prolonged cellular immune dysfunction. Use of specific agents to treat GVHD, including corticosteroids, ruxolitinib, and anti-CD52 therapies augment the risk for active TB [20]. Mortality secondary to TB pneumonia can reach up to 50%; hence, early recognition and intervention is important [19]. Findings on imaging vary from infiltrates, miliary pattern, nodules, pleural effusions, or cavitary lesions [19]. While nucleic acid testing has a sensitivity of 84% and a specificity of 99%, false-negative results may occur in the setting of recent TB exposure and low burden of mycobacteria within a specimen [19]. Culture continues to be the gold standard for diagnosis [19]. In order to prevent reactivation of TB post alloHSCT, it is important to treat for latent TB in patients with abnormal interferon-gamma release assays or tuberculin skin test with ≥5 mm induration.

The cases below (Table 1) describe the characteristics of patients with pulmonary TB along with geographic distribution. Erdstein et al. described two cases of pulmonary TB associated with cGVHD. One presented as a pleural effusion, while the other (case 1) presented as pneumonia [21]. The reported cases were from Burma, Portugal, Taiwan, Turkey, Spain, and Hong Kong.

## 6. Nontuberculous Mycobacterial Infections

Nontuberculous mycobacterial (NTM) infections are more frequent in alloHSCT recipients compared with the general population, particularly among patients with pulmonary cGVHD [1,29] Recent data report that pulmonary NTM occurred in up to 2.9% of patients who received alloHSCT [30]. Treatment and clinical appearance are typical of the general population [1].

The use of macrolides in the treatment of post-alloHSCT patients who develop bronchiolitis obliterans syndrome (BOS) is controversial [29]. More recent data suggest an association with negative outcomes, especially worse airflow-free survival and stimulation of immune cells that increase the risk of relapse [31,32]. The chronic immunocompromised state following alloHSCT, including the use of numerous immunosuppressants, is linked to a significantly greater incidence rate of NTM infection in patients receiving alloHSCT than in the general population [29]. BOS appears to be a further risk factor for the development of NTM infection, presumably reflecting an immunological condition brought on by GVHD [29].

Table 2 below summarizes all the cases noted in the literature that were able to identify the species of NTM along with patient characteristics. Cases 1, 2, and 3 were identified as part of the *M. abscessus* complex [29]. Differentiation between the subspecies of the *M. abscessus* complex, *M. abscessus* and *M. massiliense*, is important as it may affect treatment outcomes [29].

Liue et al. studied the incidence, risk factors, and survival post alloHSCT in an Asian academic center in a high endemic area. They performed a retrospective review over an 11-year span and identified 17 patients with chronic GVHD who had NTM infection. Nearly 60% of the patients had extensive GVHD with limited disease in the rest. The identification of the NTM species was performed in three patients and included *M. kansasii*, *M. avium complex*, and *M. chelonae*, while the rest were unclassified. In the *M. kansaii* case, the pathogen was isolated from spinal biopsy and a knee joint. The characteristics of the other two patients are noted below (cases 9 and 10) [24].

Cases 11 through 20 show the characteristics of patients who were treated for NTM in a University hospital in South Korea [33]. The immunosuppressants used where unknown, and it was also unknown if patients were on a macrolide part of BO management [33].

**Table 2 pathogens-12-00726-t002:** Characteristics of patients who suffered from non-tuberculous mycobacterial infections.

Patient	Age/Gender	Cancer	cGHVD/Organ Involved	Immunosuppressive	Macrolide Use	Lung Radiographic Features	NTM	Ref
Case 1	27/M	ALL	Yes/Lung (BOS)	CST + Tacrolimus	Yes	Fibrocavitary	*M. abscessus*	[29]
Case 2	47/M	Lymphoma	Yes/Lung (BOS)	CST + Tacrolimus	Yes	Nodular and Bronchiectatic	*M. abscessus*	[29]
Case 3	48/M	Lymphoma	Yes/Lung (BOS)	CST + Tacrolimus + ICS	Yes	Nodular and Bronchiectatic	*M. massiliense*	[29]
Case 4	34/M	ALL	Yes/Lung (BOS) + skin + sclera	CST	No	Cavitary nodules	*M. chelonae*	[30]
Case 5	29/M	AML	Yes/Skin	CST + Cyclosporine + Azathioprine	No	Bronchiectasisand cavitary nodules	*M. chelonae*	[34]
Case 6	33/M	CML	Yes/Skin + Liver	CST + Cyclosporine	No	Miliary pattern	*Mycobacterium fortuitum chelonae* *complex*	[35]
Case 7	40/F	CML	Yes/Unknown	Unknown	Unknown	Cavitary nodule	*Mycobacterium avium complex*	[36]
Case 8	66/F	MDS	Yes/Skin	CST + cyclosporin A + methotrexate + ECP + Ruxolitinib	Yes	Ground-glass opacities + pulmonary infiltrate	*Mycobacterium abscessus*	[37]
Case 9	49/F	AML	Yes/Skin + Mucosae	Unknown	Unknown	Unknown	*Mycobacterium avium complex*	[24]
Case 10	55/F	AML	Yes/Skin + Mucosa + Lung	Unknown	Unknown	Unknown	*Mycobacterium chelonae*	[24]
Case 11	31/F	ALL	Yes/Lung (BOS)	Yes (Unknown)	Unknown	Normal	*M. abscessus*	[33]
Case 12	34/M	Lymphoma	Yes/Lung	Yes (Unknown)	Unknown	Nonspecific pneumonia	*M. abscessus*	[33]
Case 13	21/M	AML	Yes/Lung	Yes (Unknown)	Unknown	Nonspecific pneumonia	*M. intracellulare*	[33]
Case 14	52/M	AML	Yes/Lung	Yes (Unknown)	Unknown	Bronchiectasis + nodules or infiltrate or tree-in-bud	*M. avium*	[33]
Case 15	49/M	CML	Yes/Lung (BOS)	Yes (Unknown)	Unknown	Cavitary pneumonia	*M. abscessus*	[33]
Case 16	43/M	CML	Yes/Lung (BOS)	No	Unknown	Nonspecific pneumonia	*M. avium*	[33]
Case 17	19/M	ALL	Yes/Lung (BOS)	Yes (Unknown)	Unknown	Cavitary pneumonia	*M. fortuitum*	[33]
Case 18	28/F	AML	Yes/Lung (BOS)	Yes (Unknown)	Unknown	Cavitary pneumonia	*M. intracellulare*	[33]
Case 19	40/M	AML	Yes/Lung	Yes (Unknown)	Unknown	Bronchiectasis + nodules or infiltrate or tree in bud	*M. intracellulare*	[33]
Case 20	48/M	AML	Yes/Lung (BOS)	Yes (Unknown)	Unknown	Cavitary pneumonia	*M. intracellulare*	[33]

CST = Corticosteroids.

## 7. Legionnaires’ Disease

*Legionella* is an intracellular Gram-negative bacterium of environmental origin (particularly water sources) that most commonly presents as pneumonia in an entity termed Legionnaires’ disease (LD) [38,39]. It was first described in 1976 after a fatal outbreak of respiratory illness following a American Legion convention and was attributed to contamination within the hotel’s air conditioning system [40]. More than 50 species are recognized, and the most common to cause disease in humans is *Legionella pneumophilia* serogroup 1 [40]. Legionellosis is becoming more widely acknowledged as a cause of pneumonia due to the development of more accurate diagnostic testing techniques; in the US, its prevalence increased 217% from 2000 (*n* = 1110) to 2009 (*n* = 3522) [40]. While LD can affect immunocompetent hosts, immunocompromised patients with solid tumors or hematological malignancies; solid organ transplants; or immunosuppressive medications such as tumor necrosis factor (TNF) inhibitors, corticosteroids, or antirejection medications are at increased risk [39]. Most importantly, impaired cellular immunity increases risk for severe illness due to *Legionella* [38]. *Legionella* is often acquired via community exposure either by aerosolization or aspiration of freshwater reservoirs. In addition, *Legionella* has been associated with nosocomial outbreaks, including within transplant centers [41]. Table 3 describes the cases of LD that have been found in the literature affecting the chronic GVHD population. Case 1 and case 2 were cases that relapsed after initial therapy and progressed to a lung infection [39]. Case 3 was nonresponsive to initial therapy and developed a lung abscess [42]. Case number 4 had a progressive case of legionellosis with skin involvement [40].

## 8. Nocardia

*Nocardia* is an abundant Gram-positive, aerobic bacterium found worldwide in soil, water, and decaying vegetation [43]. Pulmonary infection, generally acquired via inhalation, can present as an acute, subacute, or chronic illness. Most common clinical symptoms are fever and cough but can also manifest as non-specific night sweats, fatigue, and malaise. Radiographically, it can present as pulmonary nodules, mass-like consolidations, infiltrates, or pleural effusion [43]. Infection of the central nervous system via hematogenous spread occurs in up to 20–50% of nocardiosis [43]. Nocardiosis is rare among alloHSCT recipients, with incidence being between 0.3 and 1.7% [43]. Trimethoprim-sulfamethoxazole is commonly used as a prophylaxis against *Pneumocystis jiroveci* pneumonia (PJP) in alloHSCT recipients and is often effective in preventing infection due to *Nocardia* sp. [43]. However, despite intermittent prophylactic TMP-SMX administration, some transplant recipients develop nocardiosis, demonstrating that infection risk likely depends on both the dosage of prophylactic TMP-SMX and other factors [43]. The use of atovaquone or other alternatives to PJP prophylaxis is associated with an increased risk of nocardiosis [43]. The majority of the cases reported were not receiving TMP-SMX for PJP prophylaxis but on inhaled pentamidine, atovaquone, and dapsone (Table 4) [43,44,45,46]. Cases 3, 9, 13, 14, and 15 were on TMP-SMX [43,47,48,49]. Case 3 was on TMP-SMX prophylaxis, 80–400 mg; case 9 was on TMP-SMX, 80/400 mg; case 14 was on TMP-SMX, 20 mg/kg; and case 15 was on TMP-SMX, 160/800 mg [43,47,48,49].

## 9. Pseudomonas Aeruginosa

*Pseudomonas aeruginosa* is a Gram-negative, aerobic, rod-shaped bacterium that can be isolated from environmental reservoirs including soil, plants, and animal tissue [51,52]. Using its potent binding components, including as flagella, pili, and biofilms, this bacteria can survive on water, various surfaces, and medical equipment [51]. *P. aeruginosa* is therefore prevalent in both natural and artificial settings, such as lakes, hospitals, and domestic sink drains [51]. A variety of diseases in humans are brought on by the opportunistic bacterium *Pseudomonas aeruginosa* [51]. It is now a significant contributor to antibiotic resistance and nosocomial infections [51]. *Pseudomonas aeruginosa* is a type of opportunistic bacteria that has been linked to a number of healthcare-associated infections, such as ventilator-associated pneumonia (VAP), bloodstream infections from central lines, surgical site infections, urinary tract infections, burn wound infections, keratitis, and otitis media [51]. It is a bacterium that can quickly acquire antibiotic resistance, adapt to environmental changes, and produce a wide range of virulence factors [51]. Due in part to the infection’s capacity to defy both innate and acquired immune defenses through adhesion, colonization, and biofilm formation, as well as the production of different virulence factors that cause severe tissue damage, this pathogen can impact immunocompromised people [51]. Moreover, it contributes to illnesses with high death rates in people with cystic fibrosis, infections in newborns, cancer, and severe burns [51]. The most significant risk factors include structural lung illnesses, hematological neoplasms, transplantation, skin burns, recently used antibiotics, the presence of implants, prolonged hospitalization, and mechanical ventilation [51]. Eleven cases were found to have infections due to *P. aeruginosa* (Table 5) [36,53,54,55]. All patients received an allogeneic transplant and one patient had bronchiectasis [36,53,54,55].

## 10. Preventative and Mitigation Measures

Multiple mitigation and preventative strategies can be utilized to help decrease the risk of infection [3]. Optimized hand hygiene, avoidance of sick contacts, and development of protected hospital environments have been shown to be effective [3,12]. Regular dental care is vital as well.

Vaccination has been studied extensively in this population [3,56]. The type of HSCT, the timing of immunization after transplantation, the age at transplantation, and the presence or absence of chronic GVHD all affect immune responses and development of long-term immunity [56]. Even after receiving vaccinations, patients may still have a compromised immune system, necessitating additional safety measures to reduce the risk of contracting infections [56]. Active GVHD, its treatment, and the use of rituximab within 6 months of immunization attenuate the immunological response to vaccines [56]. Recipients of alloHSCT are recommended to receive vaccine series for *Streptococcus pneumoniae, Haemophilus influenzae* type b, SARS-CoV-2, and seasonal influenza among other others [1,3]. IVIG can be considered in alloHSCT recipients in the first 200 days post-transplant if there was profound hypogammaglobulinemia, Ig levels <400 mg/dL [1].

## Figures and Tables

**Figure 1 pathogens-12-00726-f001:**
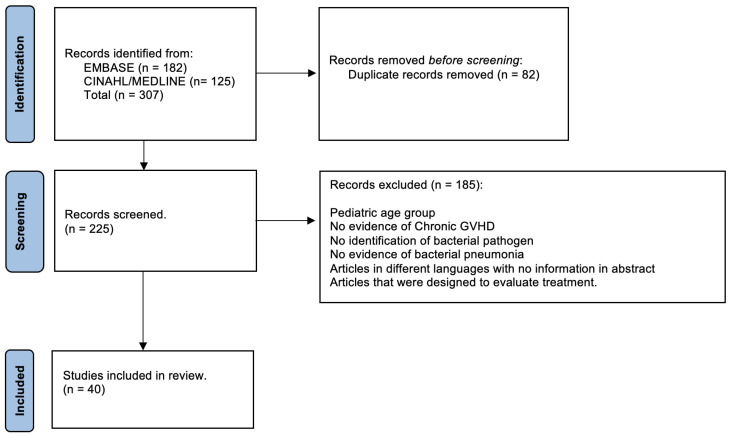
Flow chart diagram highlighting methodology.

**Table 1 pathogens-12-00726-t001:** Characteristics of patients extrapolated from the literature who suffered from mycobacterium tuberculosis.

Patient	Age/Gender	Geography	Malignancy	cGHVD/Organ Involved	Immunosuppressive	Ref
Case 1	35/F	Burma	CML	Yes/Mouth + Skin	CST + Cyclosporine	[21]
Case 2	47/M	Portugal	CML	Yes/Mucocutaneous + Ocular + Liver	CST + Cyclosporine	[22]
Case 3	47/M	Taiwan	MDS	Yes/Unknown	CST + Cyclosporine	[23]
Case 4	44/M	Taiwan	APML	Yes/Unknown	CST + Cyclosporine	[23]
Case 5	57/F	Taiwan	ALL	Yes/Eye + Skin	Unknown	[24]
Case 6	64/M	Taiwan	AML	Yes/Eye + Skin	Unknown	[24]
Case 7	58/F	Taiwan	AML	Yes/Skin + Lung + Gut + Eye + Heart	Unknown	[24]
Case 8	40/F	Taiwan	Lymphoma	Yes/Mucosa + Eye + Skin	Unknown	[24]
Case 9	33/F	Taiwan	Lymphoma	Yes/Gut + Mucosa + Skin + Lung	Unknown	[24]
Case 10	34/M	Turkey	CML	Yes/unknown	Unknown	[25]
Case 11	42/M	Spain	ANLL	Yes/Skin + Liver + GI	CST + Cyclosporine	[26]
Case 12	38/F	Hong Kong	CML	Yes/Unknown	Unknown	[27]
Case 13	40/F	Hong Kong	CML	Yes/Unknown	Unknown	[27]
Case 14	17/M	Hong Kong	AML	Yes/Unknown	Unknown	[27]
Case 15	30/F	Hong Kong	AML	Yes/Unknown	Unknown	[27]
Case 16	37/F	Hong Kong	AML	Yes/Unknown	Unknown	[27]
Case 17	54/F	Spain	CML	Yes/Unknown	Unknown	[28]
Case 18	38/M	Spain	Lymphoma	Yes/Unknown	Unknown	[28]
Case 19	39/M	Spain	ALL	Yes/Unknown	Unknown	[28]
Case 20	42/M	Spain	AML	Yes/Unknown	Unknown	[28]

CST = Corticosteroids.

**Table 3 pathogens-12-00726-t003:** Characteristics of patients who suffered from Legionnaires’ disease.

Patient	Age	Gender	cGHVD/Organ Involved	Immunosuppressive	Slow/Nonresolving LD (Lung Abscess)	Outcome	Ref
Case 1	37	F	Yes/Unknown	CST	Yes	Died (another Infectious cause)	[39]
Case 2	44	M	Yes/Unknown	CST + Calcineurin inhibitor + Cyclophosphamide	Yes	Alive	[39]
Case 3	45	M	Yes/Unknown	CST + Calcineurin inhibitor	Yes	Alive	[42]
Case 4	27	F	Yes/Skin and Bowel	CST + Tacrolimus + mycophenolate mofetil	No	Died	[40]

CST = Corticosteroids.

**Table 4 pathogens-12-00726-t004:** Characteristics of patients who suffered from Nocardiosis.

Patient	Age/Gender	Malignancy	cGHVD/Organ Involved	Immunosuppressive	Prophylaxis	Ref
Case 1	54/M	AML	Yes/Unknown	Tacrolimus + Azithromycin	Atovaquone	[43]
Case 2	52/F	AML	Yes/Unknown	CST + Tacrolimus + Mycophenolate Mofetil	Atovaquone	[43]
Case 3	52/M	Lymphoma	Yes/Unknown	CST + Tacrolimus + Budesonide	TMP-SMX	[43]
Case 4	30/F	Lymphoma	Yes/Unknown	CST + Tacrolimus + Azithromycin	Atovaquone	[43]
Case 5	63/M	CML	Yes/Unknown	CST + Rituximab	Atovaquone	[43]
Case 6	45/M	AML	Yes/Unknown	CST + Tacrolimus + Mycophenolate Mofetil	Atovaquone	[43]
Case 8	68/M	AML	Yes/Gastrointestinal tract + liver	CST + Tacrolimus	Inhaled Pentamidine	[44]
Case 9	48/M	Lymphoma	Yes/Pulmonary	CST + Cyclosporine + Inhaled CST	TMP-SMX	[47]
Case 10	50/M	AML	Yes/Skin + Mouth + Gastrointestinal	CST + Mycophenolate Mofetil	Dapsone	[45]
Case 11	17/M	Acute biphenotypic leukemia	Yes/Mouth + Skin + Liver + Lung	CST + Cyclosporine	None	[50]
Case 12	Unknown	Unknown	Yes/Unknown	Unknown	Inhaled Pentamidine	[46]
Case 13	Unknown	Unknown	Yes/Unknown	Unknown	TMP-SMX	[46]
Case 14	34/F	CML	Yes/Skin + Lung	Imatinib + IFN alpha	TMP-SMX	[48]
Case 15	48/F	CML	Yes/Skin + Gut + Liver	Unknown	TMP-SMX	[49]

CST = Corticosteroids. TMP-SMX = trimethoprim—sulfamethoxazole.

**Table 5 pathogens-12-00726-t005:** Characteristics of patients who suffered from *Pseudomonas aeruginosa* infection.

Patient	Age/Gender	Malignancy	cGHVD/Organ Involved	Bronchiectasis	Immunosuppression	Ref.
Case 1	23/M	ALL	Yes/Lung	Yes	Unknown	[53]
Case 2	49/M	AML	Yes/Lung	No	Unknown	[53]
Case 3	30/M	AML	Yes/Lung	No	CST	[54]
Case 4	36/M	ALL	Yes/Skin + Liver	No	CST + Azathioprine	[54]
Case 5	36/M	CML	Yes/Skin	No	CST	[54]
Case 6	55/F	Lymphoma	Yes/Unknown	No	Unknown	[36]
Case 7	33/M	MDS	Yes/Unknown	No	Unknown	[36]
Case 8	54/M	CLL	Yes/Unknown	No	Unknown	[36]
Case 9	51/M	Lymphoma	Yes/Unknown	No	Unknown	[36]
Case 10	43/M	AML	Yes/Unknown	No	Unknown	[36]
Case 11	25/F	AML	Yes/Unknown	No	Unknown	[55]

CST = Corticosteroids.

## Data Availability

No new data were created or analyzed in this study. Data sharing is not applicable to this article.

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
