# Peer review of "Bacterial Pathogens Causing Pneumonia Post Hematopoietic Stem Cell Transplant: The Chronic GVHD Population"

_pathogens, 2023, doi:10.3390/pathogens12050726_

Round 1

Reviewer 1 Report

The authors extensively cited case series regarding the causative pathogens of pneumonia complicating chronic GVHD. I suggest the authors consider including more of their insights on the diagnostic and treatment approaches for pneumonia in chronic GVHD. The paper would benefit from more up-to-date epidemiological information and clearer criteria for literature selection. Furthermore, it would be useful for the authors to explain why they excluded clinically significant pathogens such as Streptococcus pneumoniae and Influenza. Finally, I recommend that the authors address the inconsistencies in formatting to improve the readability of the paper.

Author Response

Reviewer 1

The authors extensively cited case series regarding the causative pathogens of pneumonia complicating chronic GVHD. I suggest the authors consider including more of their insights on the diagnostic and treatment approaches for pneumonia in chronic GVHD. The paper would benefit from more up-to-date epidemiological information and clearer criteria for literature selection. Furthermore, it would be useful for the authors to explain why they excluded clinically significant pathogens such as Streptococcus pneumoniae and Influenza. Finally, I recommend that the authors address the inconsistencies in formatting to improve the readability of the paper

Response: 

Only bacterial pathogens were included, hence why Influenza was not. Lines 81-85 discusses that Streptococcus pneumoniae and Haemophilus influenzae are well described in this population and this paper is presenting bacterial pathogens apart from these organisms.

Reviewer 2 Report

Authors are addressing how post-transplantation immune deficiency leads to bacteria-induced pneumonia specifically in the chronic GVHD in the current review article. They have some major findings in this review, but I would suggest authors should include some more latest references for compiling this review article. And I have some queries which need to be addressed for the publication.

Line 103 – Authors have used the word “high-income countries and low resourced countries”. Is it not better to simply use developed and poorly developed/developing countries?

Line 104 - 106 – How this sentence is related to M. tuberculosis incidence in high-income and low-resourced countries?

Line 106 – “The lung is usually the most affected organ”. This sentence seems abrupt due to the sudden appearance in the text. Authors should reframe the sentence

Line no. 116 – “Erdstein et. Al”. Authors should correct this mistake in the sentence.

Line no. – 102-119 – Authors have not mentioned how M. tuberculosis infection is correlated with alloHSCT therapy? The authors have included a table for this, but they should discuss such incidences in the text.

Line no. 128 – Authors should have defined BOS.

Line no. 129 – Negative outcomes of what?

Line no. 132 – Is the use of the “general” population in the sentence correct? What do authors want to say when they are referring to the population as general?

Line no. 156-174 – Authors should have elaborated on the association between alloHSCT therapy and LD. Authors have incorporated a table but it is difficult to deduce any other information from the table because scientific audiences have to go through other citations for further details.

Line no. 184 – What is the rationale behind including such bacterial infections which occur rarely during allogeneic HSCT, if it is important then why authors did not include other rare bacterial infections in the same process?

Line no. 184 – 196 – I am unable to understand what authors are trying to say with this paragraph as it only includes what is the treatment for Nocardial infection, instead, authors should address how such rare infection is linked with alloHSCT therapy.

Line no. 201, 202, 208, 209 – The name of the bacteria should be italicized

Authors should carefully use the word HSCT, alloHSCT, and allogeneic HSCT. Sometimes they have used HSCT only, at some text it is alloHSCT and at some text, it is allogeneic HSCT. It is very much confusing for a reader.

Moderate improvement is needed

Author Response

Reviewer 2

Authors are addressing how post-transplantation immune deficiency leads to bacteria-induced pneumonia specifically in the chronic GVHD in the current review article. They have some major findings in this review, but I would suggest authors should include some more latest references for compiling this review article. And I have some queries which need to be addressed for the publication.

Line 103 – Authors have used the word “high-income countries and low resourced countries”. Is it not better to simply use developed and poorly developed/developing countries? Done

The use of phrases including “developed” and “developing” are no longer favored as their orgins refer to colonialism.

Jimba M, Fujimura MS, Ong KIC. Developing country: an outdated term in The Lancet. Lancet. 2019 Sep 14;394(10202):918. doi: 10.1016/S0140-6736(19)31098-0. PMID: 31526736.

Khan T, Abimbola S, Kyobutungi C, Pai M. How we classify countries and people-and why it matters. BMJ Glob Health. 2022 Jun;7(6):e009704. doi: 10.1136/bmjgh-2022-009704. PMID: 35672117; PMCID: PMC9185389.

Line 104 - 106 – How this sentence is related to M. tuberculosis incidence in high-income and low-resourced countries?

Adjusted language without changing meaning of sentence.

Line 106 – “The lung is usually the most affected organ”. This sentence seems abrupt due to the sudden appearance in the text. Authors should reframe the sentence

Sentence reviewed and adjusted per reviewers’ recommendation.

Line no. 116 – “Erdstein et. Al”. Authors should correct this mistake in the sentence.

Mistake corrected.

Line no. – 102-119 – Authors have not mentioned how M. tuberculosis infection is correlated with alloHSCT therapy? The authors have included a table for this, but they should discuss such incidences in the text.

We added literature explaining how M. tuberculosis infection is correlated with alloHSCT.

Line no. 128 – Authors should have defined BOS.

Entity defined.

Line no. 129 – Negative outcomes of what? Negative outcomes associated with Azithromycin use were added.

Line no. 132 – Is the use of the “general” population in the sentence correct? What do authors want to say when they are referring to the population as general?

General population referred to autologous stem cell transplant, it was removed to help with adjusting language and not losing meaning of sentence.

Line no. 156-174 – Authors should have elaborated on the association between alloHSCT therapy and LD. Authors have incorporated a table but it is difficult to deduce any other information from the table because scientific audiences have to go through other citations for further details.

We added literature elaborating association between alloHSCT therapy and LD.

Line no. 184 – What is the rationale behind including such bacterial infections which occur rarely during allogeneic HSCT, if it is important then why authors did not include other rare bacterial infections in the same process?

The aim of this current review is to highlight all bacterial pathogens cauisng pneumonia in this specific entity “Chronic GVHD post stem cell transplant” hence all reported bacterial pathogens described in our search were included.

Line no. 184 – 196 – I am unable to understand what authors are trying to say with this paragraph as it only includes what is the treatment for Nocardial infection, instead, authors should address how such rare infection is linked with alloHSCT therapy.

Removed discussion about treatment and emphasized how routine PJP prophylaxis reduces incidence of nocardia

Line no. 201, 202, 208, 209 – The name of the bacteria should be italicized Done

Italicized Pseudomonas aeruginosa in lines 260-268. Bacteria names in lines 201-209 are italicized

Authors should carefully use the word HSCT, alloHSCT, and allogeneic HSCT. Sometimes they have used HSCT only, at some text it is alloHSCT and at some text, it is allogeneic HSCT. It is very much confusing for a reader.  Reviewers recommendation addressed.

Round 2

Reviewer 1 Report

The authors have provided a clarification in response to our request, which has improved the manuscript by making it clearer how the literature was selected. We appreciate the authors for addressing this issue. Additionally, we noticed a typographical error of "infant neonate adolescent" in Lane 112 that needs to be corrected. Moreover, the authors should correct "cGHVD" in Tables 1-5, which is a mistake for "GVHD." Overall, these revisions will further enhance the clarity and accuracy of the manuscript.

Author Response

Reviewer comments:

The authors have provided a clarification in response to our request, which has improved the manuscript by making it clearer how the literature was selected. We appreciate the authors for addressing this issue. Additionally, we noticed a typographical error of "infant neonate adolescent" in Lane 112 that needs to be corrected. Moreover, the authors should correct "cGHVD" in Tables 1-5, which is a mistake for "GVHD." Overall, these revisions will further enhance the clarity and accuracy of the manuscript. 

Response:

The revision the reviewer has suggested would not accurately reflect my search process.  EndNote's search feature finds words that begin with an entered word trunk.  So the search for the "infan" word trunk finds articles that include infant, infants, infancy, infantile, etc.  A search for "infant"  (the word suggested by the reviewer) would not identify instances of the word "infancy".  Similarly, a search for "neonate" will not identify instances of neonatal, and a search for "adolescent" will not identify instances of "adolescence".

We did label the tables with cGVHD and we defined it in the text as it stands for chronic GVHD and this is to confirm the study population examined in this study are those who suffered from GVHD in the chronic form.